# Structure and Antigenicity of the Porcine Astrovirus 4 Capsid Spike

**DOI:** 10.3390/v16101596

**Published:** 2024-10-11

**Authors:** Danielle J. Haley, Sarah Lanning, Kyle E. Henricson, Andre A. Mardirossian, Iyan Cirillo, Michael C. Rahe, Rebecca M. DuBois

**Affiliations:** 1Department of Biomolecular Engineering, University of California Santa Cruz, Santa Cruz, CA 95064, USA; djhaley@ucsc.edu (D.J.H.); khenricson@ucdavis.edu (K.E.H.); andrelovestennis@gmail.com (A.A.M.); iyan.cirillo@einsteinmed.edu (I.C.); 2Department of Molecular Cell and Developmental Biology, University of California Santa Cruz, Santa Cruz, CA 95064, USA; sblannin@ucsc.edu; 3Department of Population Health and Pathobiology, College of Veterinary Medicine, North Carolina State University, Raleigh, NC 27607, USA; mrahe@ncsu.edu

**Keywords:** astroviruses, virus structure, spike, porcine astrovirus, capsid

## Abstract

Porcine astrovirus 4 (PoAstV4) has been recently associated with respiratory disease in pigs. In order to understand the scope of PoAstV4 infections and to support the development of a vaccine to combat PoAstV4 disease in pigs, we designed and produced a recombinant PoAstV4 capsid spike protein for use as an antigen in serological assays and for potential future use as a vaccine antigen. Structural prediction of the full-length PoAstV4 capsid protein guided the design of the recombinant PoAstV4 capsid spike domain expression plasmid. The recombinant PoAstV4 capsid spike was expressed in *Escherichia coli*, purified by affinity and size-exclusion chromatography, and its crystal structure was determined at 1.85 Å resolution, enabling structural comparisons to other animal and human astrovirus capsid spike structures. The recombinant PoAstV4 capsid spike protein was also used as an antigen for the successful development of a serological assay to detect PoAstV4 antibodies, demonstrating that the recombinant PoAstV4 capsid spike retains antigenic epitopes found on the native PoAstV4 capsid. These studies lay a foundation for seroprevalence studies and the development of a PoAstV4 vaccine for swine.

## 1. Introduction

Astroviruses infect avian and mammalian species. Human astrovirus is a leading cause of viral gastroenteritis in children [1,2] and in rare cases causes encephalitis [3]. Despite over 90% of adults having antibodies to at least one serotype [4,5], there are no vaccines or treatments available. Astroviruses are nonenveloped, positive-sense, single-stranded RNA viruses that belong to the Astroviridae family [1,6,7] and are named after their ‘star-like’ appearance when observed by electron microscopy (EM) [8]. Their genomes vary from 6 to 8 kb and contain three ORFs called ORF1a, ORF1b, and ORF2 [1,6,7,9]. ORF1a and 1b encode non-structural polyproteins, while ORF2 encodes the viral capsid structural protein that is the target for host antibodies [9]. The capsid core domain forms the icosahedral shell around the virus genome, whereas the capsid spike domain is thought to be responsible for the attachment and entry of the virus [10,11,12]. The capsid spike domain is the primary target of serum antibodies, is the target of all known neutralizing monoclonal antibodies, and is being evaluated as a subunit vaccine antigen [10,11,12,13,14,15].

Porcine astrovirus (PoAstV) was first detected by EM in the feces of piglets with diarrhea in 1980 [16,17]. There are five known genotypes of PoAstV [18,19], which are thought to be more closely related to other species than to each other [20,21]. This divergence among genotypes suggests a different ancestral origin of PoAstVs. PoAstVs have been detected across the globe including South Africa [7,22] Canada [21], China [23], Colombia [7,24], and Chile [24], and all five genotypes are present in the US with high incidence [17,20]. Therefore, PoAstV is thought to have a wide geographical distribution [7] and to be endemic in commercial swine in the US [25]. One study of fecal samples from 509 pigs from 255 farms across 19 US states showed PoAstV4 had the highest prevalence [20] at 62% (317/509), and 64% (326/509) had at least one of the PoAstV genotypes. Multiple astroviruses have been detected in a single pig at once, [7,17,20,21], which could provide an opportunity for recombination to occur and lead to the emergence of new strains [7].

Multiple studies have connected PoAstV to a range of disease manifestations, with the virus frequently detected in the feces of pigs displaying diarrheal symptoms as well as asymptomatic pigs [7,17,21]. PoAstV5 is a cause of clinical enteritis [26], while PoAstV3 has been identified and characterized in the central nervous system of pigs with neurologic signs and nonsuppurative polioencephalomyelitis [27]. Additionally, PoAstV4 has been identified in nasal samples from pigs with respiratory disease [17].

Recently, researchers at Iowa State University investigated cases of bronchitis and/or tracheitis in pigs where PCR results were negative for influenza virus and other known causes of respiratory virus infection in pigs [25]. Next-generation sequencing revealed reads of PoAstV4, leading to the hypothesis that the respiratory disease in these pigs was associated with PoAstV4. In a retrospective study of cases of tracheitis and/or bronchitis of unknown etiology, RNA in situ hybridization (ISH) was used to detect PoAstV4 RNA in airway epithelium (trachea, bronchi, or bronchioles), revealing PoAstV4 RNA in 73% (85/117) of cases [25]. This reveals that PoAstV4 is strongly associated with lesions of epitheliotropic viral infection in young pigs with clinical respiratory disease.

To further understand PoAstV4 infection and prevalence, we generated a recombinant PoAstV4 capsid spike protein for structural and antigenic studies. We determined the crystal structure of the PoAstV4 capsid spike at 1.85 Å resolution, validating its three-dimensional folding and enabling structural comparisons to the capsid spikes from other astroviruses. We also used the PoAstV4 capsid spike protein as an antigen for the development of an enzyme-linked immunosorbent assay and demonstrated that the recombinant PoAstV4 capsid spike retains antigenic epitopes found on the native PoAstV4 capsid. These studies provide a basis for future studies to understand the prevalence of PoAstV4 in pigs and to develop a PoAstV4 capsid spike subunit vaccine to prevent PoAstV4 disease.

## 2. Materials and Methods

### 2.1. Phylogenetic Analysis of Astroviruses with MEGA X

Amino acid sequences of the full-length astrovirus capsid ORF2 were aligned using MUSCLE (EMBL-EBI, Heidelberg, Germany) [28]. The following ORF2 sequences were used: human astrovirus 1, GenBank #AAC34717.1; human astrovirus 2, UniProt #Q82446.1; human astrovirus 3, UniProt #Q9WFZ0.1; human astrovirus 4, UniProt #Q3ZN05.1; human astrovirus 5, UniProt #Q4TWH7.1; human astrovirus 6, UniProt #Q67815.1; human astrovirus 7, UniProt #Q96818.2; human astrovirus 8, UniProt #Q9IFX1.2; human astrovirus VA1, GenBank #YP_003090288.1; human astrovirus VA2, NCBI #ACX83591.2; human astrovirus VA3, NCBI #YP_006905860.1; human astrovirus VA4, NCBI #YP_006905857.1; human astrovirus VA5; human astrovirus MLB1, NCBI #YP_002290968.1; human astrovirus MLB2, GenBank #YP_004934010.1; human astrovirus MLB3, GenBank #YP_006905854.1; murine astrovirus, GenBank #QBQ83077.1; turkey astrovirus 1, UniProt #Q9JH68; turkey astrovirus 2, UniProt #Q9Q3G5; turkey astrovirus 3, GenBank #AAV37187.1; porcine astrovirus 1, GenBank #UZG75482.1; porcine astrovirus 2, GenBank #AZB49326.1; porcine astrovirus 3, NCBI #YP_007003832.1; porcine astrovirus 4, GenBank #AMN16564; porcine astrovirus 4, GenBank #AMN16570.1; porcine astrovirus 4, GenBank #PP806170.1; porcine astrovirus 5, GenBank #UXD79156.1; mink astrovirus (mamastrovirus 10), GenBank #AAO32083.1; ovine astrovirus (mamastrovirus 13), GenBank #QDP38704.1; bovine astrovirus (neuro), GenBank #ANS5716.1; bovine astrovirus (gastrointestinal), GenBank #BAS29642.1; sea lion astrovirus (mamastrovirus 4), GenBank #QPD02148.1; wild boar astrovirus, NCBI #YP_005271209.1; bat astrovirus, GenBank #QOR29563.1. The evolutionary history of astrovirus species was inferred by using the maximum-likelihood method and a Jones–Taylor–Thornton (JTT) matrix-based model [29]. The tree shown has the highest log likelihood (−43,729.83). The percentage of trees in which the associated taxa clustered together is shown next to the branches. Initial tree(s) for the heuristic search were obtained automatically by applying neighbor-joining and BioNJ algorithms to a matrix of pairwise distances estimated using the JTT model, and then selecting the topology with superior log likelihood value. The tree is drawn to scale, with branch lengths measured in the number of substitutions per site. The tree was rooted using the turkey astroviruses as an outgroup. There were 34 amino acid sequences in this dataset and 1046 sites. Evolutionary analyses were conducted in MEGA X [30].

### 2.2. Comparison of Pairwise Identity between PoAstV4 Spike and HAstV Spikes

MUSCLE (EMBL-EBI) alignment was used to perform pairwise identity between astrovirus spike sequences. The same accession numbers utilized in the phylogenetic analysis section were used for the human astrovirus spikes, murine astrovirus spike, and for the porcine astrovirus 1, 2, 3, and 5 spikes. AlphaFold3 [31] was used to predict the spike domain from the full-length capsid for PoAstV2, PoAstV3, and PoAstV5 and MUSCLE alignment of the full-length capsid was used to predict the spike domain for PoAstV4. The subsequent amino acids corresponding to spike domains in the ORF2 capsid protein were used: PoAstV1, 420–669; PoAstV2, 409–648; PoAstV3, 480–736; PoAstV4 (GenBank #PP806170.1), 420–655; PoAstV4 (GenBank #QEQ91926.1), 421–662; PoAstV4 (GenBank #BAX00239.1), 419–652; PoAstV4 (GenBank #BAX00233.1), 419–652; PoAstV4 (GenBank #UHY36868), 419–651; PoAstV4 (GenBank #UXM19193.1), 420–652; PoAstV4 (GenBank #AMN16570.1), 420–653; PoAstV4 (GenBank #AMN16564), 419–651; PoAstV4 (GenBank #QWT72249.1), 465–705; PoAstV4 (GenBank #QDZ38040.1), 419–649; PoAstV5, 470–689; HAstV1, 431–644; HAstV2,429–644; HAstV3, 432–645; HAstV4, 430–644; HAstV5, 429–641; HAstV6, 430–642; HAstV7, 431–644; HAstV8, 490–705; HAstV-MLB1, 420–646; HAstV-MLB2, 417–643; HAstV-MLB3, 417–643; HAstV-VA1, 408–682; HAstV-VA2, 404–688; HAstV-VA3, 388–691; HAstV-VA4, 408–685; HAstV-VA5, 406–678, MuAstV, 428–676.

### 2.3. Design and Production of Recombinant PoAstV4 Capsid Spike Protein

The full-length PoAstV4 capsid protein sequence derived from an infected pig in 2022 was retrieved from GenBank (GenBank #PP806170.1) and its structure was predicted using the AlphaFold2 server ColabFold [32]. The predicted structure was used to delineate the termini of the PoAstV4 capsid spike domain as amino acids (420–655). An *E. coli*-codon-optimized synthetic gene encoding the PoAstV4 spike was cloned into the pET52b expression plasmid in-frame with an N-terminal methionine and a C-terminal thrombin protease cleavage site and a 10×-histidine tag by GenScript. The plasmid was transformed into T7 Express *E. coli* (New England Biolabs, Ipswich, MA, USA) and grown at 37 °C in Luria Broth with 50 μg/mL ampicillin. Expression of recombinant PoAstV4 capsid spike was induced with 1 mM isopropyl-β-d-thiogalactopyranoside (IPTG), and the cultures were shaken overnight at 18 °C. *E. coli* cultures were centrifuged, and pellets were resuspended in Buffer A (20 mM Tris-HCl pH 8.0, 500 mM NaCl, 20 mM imidazole) containing 1× EDTA-free protease inhibitors (Millipore, Burlington, MA, USA), benzonase (Millipore, Burlington, MA, USA), and 2.5 mM MgCl2. *E. coli* was lysed with ultrasonication, and the lysate was centrifuged at 40,000× *g* for 30 min. The supernatant was 0.22 μm filtered and then incubated with TALON beads prewashed with Buffer A for 1 h at 4 °C with rotation. The beads were washed 12 times with Buffer A, and recombinant PoAstV4 spike was eluted in Buffer B (20 mM Tris-HCl pH 8.0, 500 mM NaCl, 500 mM imidazole). Approximately 73 mg of recombinant PoAstV4 spike protein was obtained from a 6 L expression. The 10×-histidine tag was removed from a portion of the PoAstV4 spike protein by incubation with bovine thrombin protease (Millipore 60-516-01KU, Burlington, MA, USA) overnight at 4 °C during dialysis into TBS (10 mM Tris–HCl pH 8.0, 150 mM NaCl). Remaining PoAstV4 spike protein with 10×-histidine tag intact was buffer exchanged into TBS using a desalting column (Cytiva #170851, Marlborough, MA, USA). The PoAstV4 spike proteins with or without the 10×-hisidine tag were purified further by size exclusion chromatography using a Superdex 200 16/600 column in TBS pH 8.0.

### 2.4. Structural Determination of PoAstV4 Spike

Purified PoAstV4 spike was concentrated to 23 mg/mL in TBS. PoAstV4 spike protein crystals were grown in 2 µL drops consisting of a 1:1 ratio of protein solution to well solution containing 0.2 M calcium acetate hydrate, 0.1 M Tris-HCl pH 7, and 12% PEG 3000, using hanging drop vapor diffusion at 22 °C. One crystal was transferred into a cryoprotectant solution containing the well solution with 25% PEG 400 before being flash-frozen into liquid nitrogen. The Advanced Light Source Beamline 5.0.1 was used to collect X-ray diffraction data with a wavelength of 0.97 Å at cryogenic temperatures. Diffraction data were highly anisotropic and a number of frames (75/720) had to be omitted due to poorly defined spots. This dataset was processed and scaled with DIALS (ccp4i2 8.0.016, Didcot, UK) using a low-resolution cutoff of 40 Å and a high-resolution cutoff of 1.85 Å. A model generated by AlphaFold2/ColabFold [31] was trimmed to remove residues 432–435, 625–628, and 640–655, and this model was used for molecular replacement with Phaser. The molecular replacement phase data were used for the first round of refinement to build an initial model, which was modeled manually using Coot (0.9.8.92, Cambridge, UK) [33] and refined in Phenix [34] to create an improved model in chain A. Chain A was used for molecular replacement on the other chains, and AutoBuild was then applied to correct incorrectly placed residues. The model was polished using Coot (0.9.8.92, Cambridge, UK) [33] and refined in Phenix [34] to generate the final structure. Calcium ions were modeled, as they fit the CheckMyMetal (Charlotttesville, VA, USA) [35] parameters best in comparison to other ions of similar size or charge (Mg, K, Mn, Na) that were modeled and tested for coordination chemistry, agreement of experimental B-factors, occupancy, and the metal binding environmental motif. There was calcium acetate in the crystallography condition, which is the likely source of the calcium ion.

### 2.5. Development of an Enzyme-Linked Immunosorbent Assay to Evaluate PoAstV Antibodies in Pig Sera

Serum samples from pigs with known or suspected exposure to PoAstV4 were used as presumed seropositive serum samples. Serum from cesarean-derived, colostrum-deprived (CDCD) pigs were used as presumed seronegative serum samples [36]. A ninety-six well medium-binding ELISA plate (Corning #9017) was coated with 50 µL of 10 µg/mL purified PoAstV4 spike in phosphate-buffered saline (PBS), PBS alone as a “no antigen” control, covered with microplate sealing tape (Corning #6575) and incubated overnight at 4 °C. The plate was washed three times (200 μL each) with PBS-T (PBS + 0.1% Tween). Then, 200 μL of blocking solution (PBS-T + 5% milk) was added to all wells of the plate and incubated for 2 h at room temperature. After the incubation, the blocking solution was thrown off the plate and tapped on a Kimwipe to dry. Next, 80 μL of blocking buffer was added to all wells and an extra 64 μL was added to Row A. Then, 16 μL of a prediluted 1:10 sera sample was added to row A, making an initial 1:100 dilution on the plate. Each serum sample was added in triplicate, with three columns dedicated to an individual sample. A multichannel pipette was used to pipette up and down 4–6 times in row A and to transfer 80 μL to row B. This was repeated through row G (no sera in the last row), and the last 80 μL were discarded. The plate was incubated for 2 h at room temperature, and then washed 3 times with 200 μL PBS-T. Next, 50 μL of Anti-swine IgG-HRP secondary antibody (Jackson ImmunoResearch Laboratories #114-035-003, West Grove, PA, USA) diluted at 1:10,000 in PBS-T + 1% milk was added to each well, and the plate was incubated for 1 h at room temperature. The plate was then washed three times with 200 μL PBS-T. To develop the plate, 100 μL of TMB substrate (Sigma #T0440, St. Louis, MO, USA) was added to all wells and incubated for 9 min at room temperature, followed immediately by quenching with 100 μL 1 N H_2_SO_4_. The 450 nm absorbance in each well was measured with a plate reader.

## 3. Results

### 3.1. Delineation and Production of Recombinant PoAstV4 Capsid Spike Domain

We investigated the capsid protein derived from the sequence of PoAstV4 isolated from a pig with respiratory disease [25]. This PoAstV4 sequence clusters with other known PoAstV4 capsid sequences on a phylogenetic tree, but it does not cluster with other PoAstV capsid sequences (Figure 1A). The protein sequence of the PoAstV4 capsid spike has a low sequence identity to the capsid spikes of other astrovirus species (Figure 1B). Therefore, we could not determine the spike domain boundaries based on sequence alignment alone, and instead determined the spike domain boundaries by predicting the structure of the full PoAstV4 capsid protein using AlphaFold2 [32]. This PoAstV4 capsid spike sequence showed 39–94% identity to other PoAstV4 capsid spikes (Figure 1B). We then designed a PoAstV4 spike construct containing residues 420 to 655 from the full-length capsid protein of PoAstV4 (Figure 1C). The recombinant PoAstV4 spike protein expressed in *E. coli* resulted in high yields of soluble protein (>12 mg/liter *E. coli*). The calculated molecular weight of the PoAstV4 spike is 27.8 kD (+1.6 kD cleavable histidine affinity tag), and this was confirmed using SDS-PAGE analysis of purified PoAstV4 spike (Figure 2A, Appendix A). Size exclusion chromatography showed purified PoAstV4 spike protein eluted at the approximate size of a homodimer (Figure 2B), which suggests that the recombinant protein is folded correctly.

### 3.2. Crystal Structure of PoAstV4 Spike and Similarity to Other Astroviruses

To further substantiate the folding of the recombinant PoAstV4 spike, we used X-ray crystallography to determine the structure of the PoAstV4 spike to 1.85 Å resolution (Figure 3, Table 1). A trimmed AlphaFold2 model was used for molecular replacement. The high R factors are likely due to the highly anisotropic data; however, the electron density maps are clear (Figure 3B). Structural alignment of the crystal structure of the PoAstV4 spike dimer with an AlphaFold2 predicted model using TM-align [37] reveals an RMSD of 0.72 Å and a TM alignment score of 0.886 (Figure 3C), whereas an alignment with an AlphaFold3 [30] predicted model reveals an RMSD of 0.49 Å and a TM alignment score of 0.992 (Figure 3D), demonstrating the accuracy of these structure prediction programs and improvements from AlphaFold2 to AlphaFold3.

The PoAstV4 capsid spike crystal structure reveals a homodimeric protein formed of mainly beta-strands (Figure 3), similar to other animal and human astrovirus capsid spikes, and consistent with in-solution studies with size-exclusion chromatography (Figure 2B). The structure confirms that the predicted PoAstV4 spike residues 420 to 655 form the structural domain. Interface analysis using the PDBePISA server reveals 4660 Å buried at the dimer interface, with 64–65 interacting residues in each chain.

Compared to other experimentally solved astrovirus spike structures [12,13,38,39,40], the PoAstV4 spike has the highest structural similarity to murine astrovirus (MuAstV) (Figure 4), as seen visually with a notable beta sheet cleft at the top of both spikes and by their high TM alignment score [37] of 0.809 (Figure 4). This TM alignment score between PoAstV4 and MuAstV spike is the highest compared to other astrovirus spikes assessed (Figure 4).

### 3.3. Development of an ELISA to Detect PoAstV4 Antibodies in Pig Sera

To determine if the recombinant PoAstV4 spike can be used as an antigen to detect PoAstV antibodies, we developed an enzyme-linked immunosorbent assay (ELISA). Serum samples were collected from pigs in Iowa with suspected exposure during an outbreak of PoAstV4, making them presumed to be seropositive. A strong dose-dependent IgG reactivity towards the PoAstV4 spike antigen was observed in sera from presumed seropositive PoAstV4 adult pigs (Figure 5A). Sera from presumed seronegative cesarean-derived, colostrum-deprived (CDCD) pigs, which do not receive maternal antibodies, exhibited minimal reactivity (450 nm absorbance values < 0.187) and all sera had a low signal in the negative control plate not coated with antigen (<0.197 nm at 450 nm) (Figure 5B), validating the ELISA specificity.

## 4. Discussion

PoAstVs represent a significant concern within swine populations, posing health risks to individual pigs in addition to financial burdens on swine farms. While PoAstVs are traditionally associated with diarrheal disease, PoAstV4 has recently been associated with respiratory disease in young pigs [27]. Here, we investigated the structural and antigenic features of the PoAstV4 capsid spike.

Phylogenetic analyses highlight the divergence of PoAstV lineages, indicating likely distinct ancestral origins. In particular, the PoAstV4 capsid exhibits sequence similarity with the murine astrovirus capsid and sea lion astrovirus capsid more than with other known PoAstV capsids, with an amino acid sequence identity of ~40% and ~50%, respectively, and clustering together in phylogenetic trees. The structural studies here further support this evolutionary relationship, with the PoAstV4 spike sharing structural similarities, and a higher alignment score with the murine astrovirus spike structure than with the PoAstV1 spike structure. Future structural studies on other PoAstV spikes may further illuminate their evolutionary relationship with other mammalian astroviruses.

In human astroviruses, the capsid spike is the primary target of host antibodies [10,11,12,13,14,15]. As such, recombinant human astrovirus capsid spikes have been used as antigens in immunoassays for evaluating seroprevalence in human populations [4]. To determine if the PoAstV4 spike is also targeted by host antibodies, we developed an ELISA to detect PoAstV spike-reactive IgG in pig sera. We observed dose-dependent reactivity for the PoAstV spike antigen by two presumed seropositive samples, suggesting previous infection by PoAstV4.

A limitation of this study is that we did not have serum samples from pigs who were confirmed by RT-PCR to be infected with PoAstV4. However, control experiments with CDCD pig sera, or with plates not coated with PoAstV4 spike antigen, confirmed the specificity of the ELISA. Future studies with sera from pigs before and after RT-PCR-confirmed PoAstV4 infection will further validate the ELISA. It is important to note that it is highly unlikely that the reactivity by the presumed seropositive pig sera was due to antibodies from previous infection by another PoAstV, as there is low or no sequence identity between the PoAstV4 spike and other PoAstV spikes (11.5–19.6% identity, Figure 1B). Even antibodies to classical human astrovirus 1–8 spikes, which have much higher sequence identities of 40–90%, do not cross-react, defining the classical human astroviruses as serotypes [5]. Overall, the ELISA developed in this study allows for the detection of PoAstV4 antibodies in pig sera, enabling future serological studies to understand the antibody prevalence and infer PoAstV4 exposure in pigs, to understand seroconversion after infection, and to evaluate seroconversion during vaccine development.

## 5. Patents

The authors have submitted a provisional patent application based on this work.

## Figures and Tables

**Figure 1 viruses-16-01596-f001:**
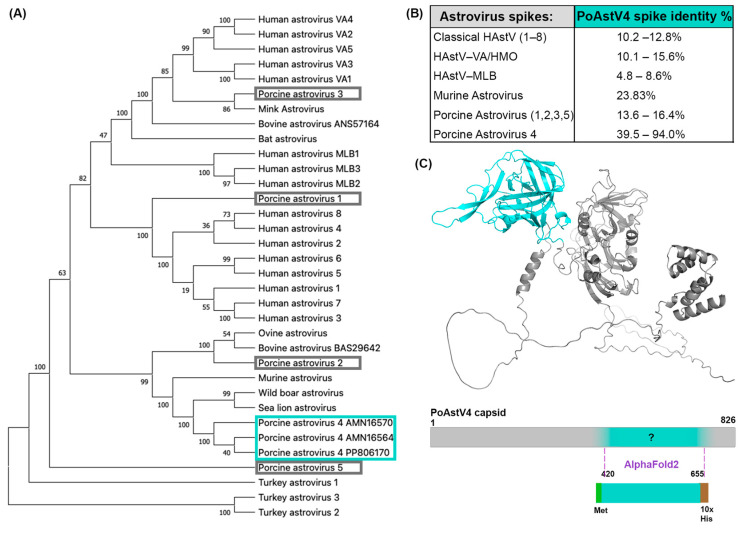
Predicting the boundaries of the PoAstV4 capsid spike domain: (**A**) Phylogenetic analysis of human and animal astroviruses was performed using MEGA X with a MUSCLE (EMBL-EBI) alignment of full-length ORF2 capsid protein sequences and the maximum-likelihood method and a JTT matrix-based model. The tree with the highest log likelihood (−43,729.83) is shown. The PoAstV4 and BoAstV capsid sequences are labeled with their accession number. The PoAstV4 sequences are boxed in cyan and PoAstV1, 2, 3, 5 sequences are boxed in gray. (**B**) Pairwise amino acid sequence identity between PoAstV4 spike and other astrovirus spikes, generated using MUSCLE (EMBL-EBI) alignment. (**C**) Top: AlphaFold2 prediction of the full-length PoAstV4 capsid, with the predicted spike domain colored in cyan, and the rest of the sequence in gray. Bottom: Design of the recombinant PoAstV4 spike expression construct showing the predicted residues for the PoAstV4 spike domain.

**Figure 2 viruses-16-01596-f002:**
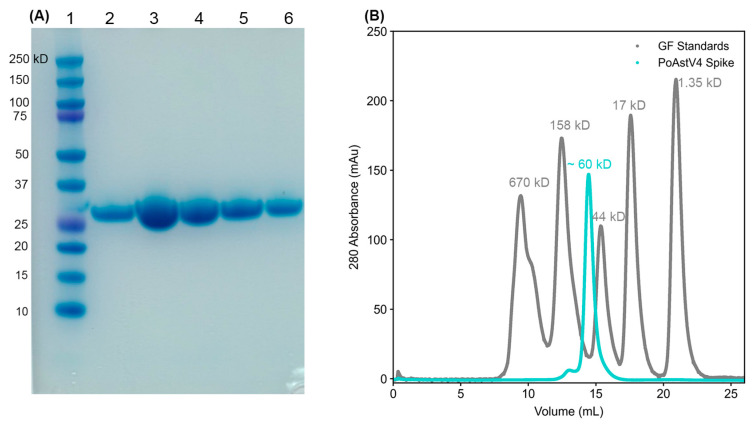
Purification of recombinant PoAstV4 capsid spike protein: (**A**) SDS-PAGE of affinity purification elution fractions of recombinant PoAstV4 capsid spike protein. Lane 1, BioRad Precision Plus molecular weight markers; Lane 2, final wash; Lane 3–6, purified PoAstV4 spike elutions. (**B**) Size exclusion chromatography traces on a Superdex 200 column of PoAstV4 spike in cyan and BioRad gel filtration standards in gray.

**Figure 3 viruses-16-01596-f003:**
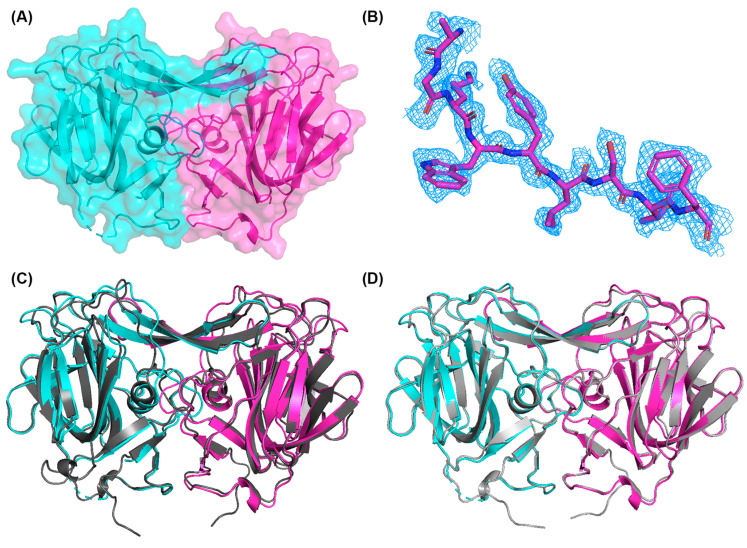
Structure of the porcine astrovirus 4 capsid spike protein: (**A**) PoAstV4 spike dimer, with individual protomers colored in magenta and cyan. (**B**) Electron density maps contoured at 1σ around residues 591–599. (**C**) Overlay of the AlphaFold2 predicted model (dark gray) with the experimentally determined crystal structure. (**D**) Overlay of the AlphaFold3 [27] predicted model (light gray) with the crystal structure. Figures were generated in PyMOL.

**Figure 4 viruses-16-01596-f004:**
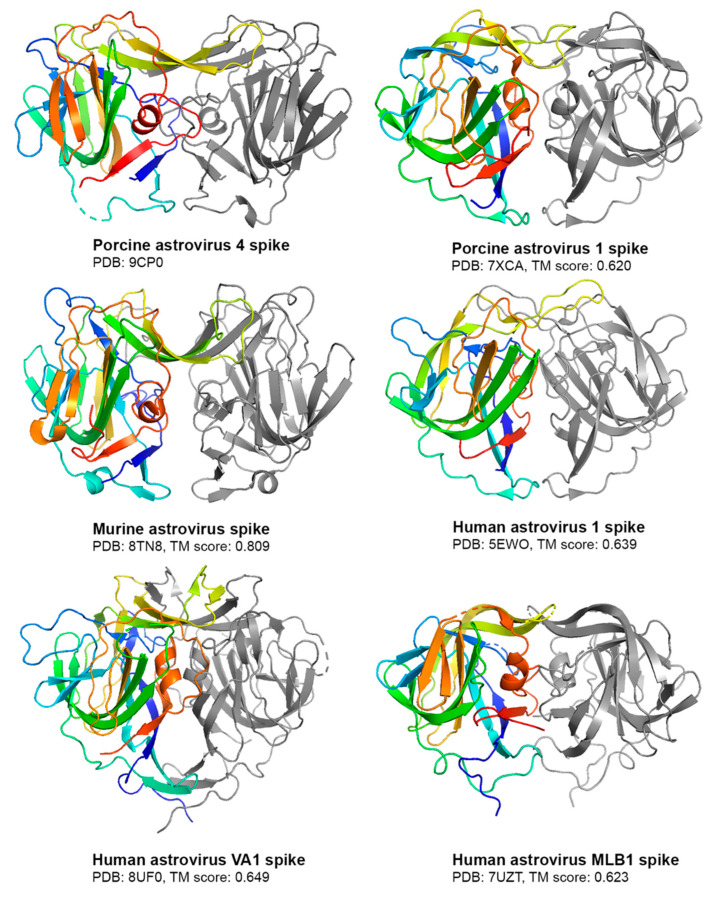
Comparison of the PoAstV4 spike to other astrovirus spikes. Structures are shown as cartoons, with one protomer colored rainbow and the other protomer colored gray. PDB codes are noted. TM alignment scores between the PoAstV4 spike and the respective astrovirus spike are reported below its structure. Figures were generated in PyMOL.

**Figure 5 viruses-16-01596-f005:**
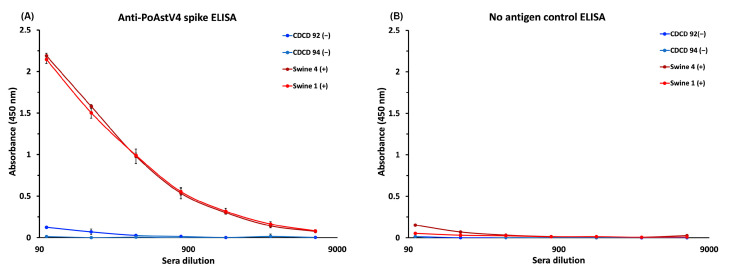
Anti-PoAstV4 spike IgG ELISA: (**A**) ELISA data showing a dose-dependent response towards recombinant PoAstV4 spike antigen in two presumed seropositive (+) pig serum samples. Low-no reactivity was observed in the presumed seronegative (−) pig serum samples from two CDCD piglets. Each sample was measured in triplicate, with the average reported, and error bars represent the standard deviation. (**B**) A matching negative control ELISA using an ELISA plate not coated with an antigen showed low or no reactivity (<0.197 absorbance at 450 nm), supporting the specificity of the ELISA.

**Table 1 viruses-16-01596-t001:** Data collection and refinement statistics.

PoAstV4 Spike	
**PDB entry:**	9CP0
**Data collection:**	
Wavelength (Å)	0.97
Space Group	P1
Cell dimensions:	
a, b, c (Å)	46.39, 66.21, 87.75
α, β, ɣ (°)	74.54, 81.67, 78.61
Resolution (Å)	38.33–1.85 (1.91–1.85) *
Rmerge	0.144 (0.474)
I/σl	8.1 (2.1)
Completeness	96.1% (93.9%)
Multiplicity	3.2 (3.4)
CC_1/2_	0.984 (0.814)
**Refinement:**	
Resolution	39.78–1.85
No. reflections for refinement	80,308
No. reflections for Rfree	1998
Rwork/Rfree	0.316/0.362
No. atoms	7412
Protein	7128
Ligand/ion	4
Water	280
B factors (Å^2^):	15.29
Protein	15.30
Ligand/ion	16.29
Water	14.99
RMSD:	
Bond lengths (Å)	0.005
Bond angles (°)	0.74
Ramachandran statistics:	
Favored (%)	96.77
Allowed (%)	3.23
Outliers (%)	0.00

* Values for the highest resolution shell are shown in parentheses.

## Data Availability

The coordinates and structure factors for the PoAstV4 capsid spike crystal structure have been deposited with the Protein Data Bank (PDB; https://www.rcsb.org accessed on 17 July 2024) under PDB entry code 9CP0.

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
