# Peer review of "Structure and Antigenicity of the Porcine Astrovirus 4 Capsid Spike"

_viruses, 2024, doi:10.3390/v16101596_

Round 1
Reviewer 1 Report
Comments and Suggestions for Authors
The article "Structure and antigenicity of the porcine astrovirus 4 capsid spike" by Haley et al is a well written, concise manuscript describing important structural characteristic of the capsid of PoAstV4; and enteric swine virus, recently described as a putative cause of respiratory disease in pigs. The authors clearly convey their objective, methodology and conclusions.
Major comments: The phylogenic analysis is based in few astrovirus strains, The viral landscape of both, mamastroviruses and avastroviruses is very vast, with hundreds of strains with complete CD for capsid protein.
Thus, why the selection for phylogenic analysis only included few selected strains, why some mamastrovirus were excluded? i.e. bovine, ovine, marine and exotic animals strains? Please would you better provide the rationale for this on the methods section.
Authors should explain well the rationale for this decision as it markedly hinders the design and conclusions. I do understand that a comprehensive analysis is cumbersome, but at least the the rationale should be clearly stated.
Minor suggestions:
- The sentence starting at line 254 and ending at 256 should be located at the conclusion-discussion section as it is an interpretation of the results.
- Table 1 should be rescaled as font and layout may be minimized for better readability. If no possible it should be provided as supplemental material.
Author Response
Reviewer 1:
The article "Structure and antigenicity of the porcine astrovirus 4 capsid spike" by Haley et al is a well written, concise manuscript describing important structural characteristic of the capsid of PoAstV4; and enteric swine virus, recently described as a putative cause of respiratory disease in pigs. The authors clearly convey their objective, methodology and conclusions.
Response:
We thank the reviewer for this positive review.
Major comment 1:
The phylogenic analysis is based in few astrovirus strains, The viral landscape of both, mamastroviruses and avastroviruses is very vast, with hundreds of strains with complete CD for capsid protein. Thus, why the selection for phylogenic analysis only included few selected strains, why some mamastrovirus were excluded? i.e. bovine, ovine, marine and exotic animals strains? Please would you better provide the rationale for this on the methods section. Authors should explain well the rationale for this decision as it markedly hinders the design and conclusions. I do understand that a comprehensive analysis is cumbersome, but at least the the rationale should be clearly stated.
Response:
We agree and have added wild boar, sea lion, bovine, and bat astrovirus sequences to Figure 1A create a more comprehensive phylogenetic tree for mamastroviruses. However, we note that the goal was not necessarily to have a comprehensive tree but to show where PoAstV4 resides compared to other PoAstVs and compared to human and mouse astroviruses, for which we have capsid spike structures for. The other aim of this tree is to show how PoAstV genotypes do not cluster together on a phylogenetic tree. We have clarified these points in lines 212-215.
Minor suggestion 1:
- The sentence starting at line 254 and ending at 256 should be located at the conclusion-discussion section as it is an interpretation of the results.
Response:
We agree. The sentence “At first, it may be surprising that the PoAstV1 spike is not the most structurally similar to PoAstV4 spike, however phylogenetic analyses suggests that PoAstV4 is more evolutionarily related to MuAstV (Figure 1A).” has been removed as we agree it is an interpretation of results, and already have a similar statement in the discussion section at lines 314-316.
Minor suggestion 2:
- Table 1 should be rescaled as font and layout may be minimized for better readability. If no possible it should be provided as supplemental material.
Response:
We agree. Table 1 has been reformatted to have less spacing between lines.
Reviewer 2 Report
Comments and Suggestions for Authors
Manuscript by Haley et al (viruses-3239320) entitled “Structure and antigenicity of the porcine astrovirus 4 capsid spike” described structural determination of viral capsid spike of porcine astrovirus 4 (PoAstV4) and examination of its antigenicity using the recombinant spike protein. Structural examination of PoAstV4, which apparently associated with clinical respiratory disease in young pig, could guide a way to effective prevention and surveillance of the PoAstV4 infection and disease. Contrast to the superb work on structural analysis of PoAstV4 spike, serological analysis using recombinant spike protein to survey presence of anti-PoAstV4 in pooled sera was somewhat incomplete. The authors could provide information, such as more detailed phylogenetic analysis, on PoAstV4 sequences utilized to generate recombinant spike (GenBank#PP806170.1) if it was representative to currently circulating PoAstV4 strain. More practical solution would be that using recombinant protein of other astrovirus spikes as control antigens to the assay in Fig. 4. This reviewer felt that this manuscript would need modifications for publication in Viruses.
Author Response
Reviewer 2:
Manuscript by Haley et al (viruses-3239320) entitled “Structure and antigenicity of the porcine astrovirus 4 capsid spike” described structural determination of viral capsid spike of porcine astrovirus 4 (PoAstV4) and examination of its antigenicity using the recombinant spike protein. Structural examination of PoAstV4, which apparently associated with clinical respiratory disease in young pig, could guide a way to effective prevention and surveillance of the PoAstV4 infection and disease. Contrast to the superb work on structural analysis of PoAstV4 spike, serological analysis using recombinant spike protein to survey presence of anti-PoAstV4 in pooled sera was somewhat incomplete. The authors could provide information, such as more detailed phylogenetic analysis, on PoAstV4 sequences utilized to generate recombinant spike (GenBank#PP806170.1) if it was representative to currently circulating PoAstV4 strain.
Response:
We thank the reviewer for this positive review and insightful feedback. We have aligned all ten publicly-available full-length PoAstV4 capsid sequences at their spike region and reported their sequence identity in an updated Figure 1B. We selected a PoAstV4 strain for further study because it was isolated from a pig with respiratory disease, and we now state this at lines 212-213. We also provide sequence identity information for this PoAstV4 strain spike compared to other PoAstV4 spike sequences at lines 219-220 and in Figure 1B. Furthermore, this PoAstV4 strain for further study because it aligns with the location and collection date of the presumed positive sera, as stated at lines 187-189.
More practical solution would be that using recombinant protein of other astrovirus spikes as control antigens to the assay in Fig. 4. This reviewer felt that this manuscript would need modifications for publication in Viruses.
Response:
We tried this experiment plating with human astrovirus 1 capsid spike or murine astrovirus capsid spike as negative control antigens, but the seropositive pig serum samples displayed some low-moderate reactivity. Upon further thought, we realized there is no way to know if these pigs have been exposed to other animal and/or human astroviruses. Thus, we don’t believe that use of other astrovirus spikes make a relevant negative control. Instead, we would like to emphasize that the plates are blocked with milk proteins that serve as a negative control antigen, for which we observe very low signal, as seen in Figure 5B. Also, we reiterate the controls used in the ELISA at lines 330-332, and we acknowledge limitations of the ELISA at lines 329-330.